# Stein Variational Gradient Descent with Matrix-Valued Kernels

Dilin Wang[*]    Ziyang Tang[*]    Chandrajit Bajaj    Qiang Liu
Department of Computer Science, UT Austin
{dilin, ztang, bajaj, lqiang}@cs.utexas.edu

## Abstract

Stein variational gradient descent (SVGD) is a particle-based inference algorithm that leverages gradient information for efficient approximate inference. In this work, we enhance SVGD by leveraging preconditioning matrices, such as the Hessian and Fisher information matrix, to incorporate geometric information into SVGD updates. We achieve this by presenting a generalization of SVGD that replaces the *scalar-valued* kernels in vanilla SVGD with more general *matrix-valued* kernels. This yields a significant extension of SVGD, and more importantly, allows us to flexibly incorporate various preconditioning matrices to accelerate the exploration in the probability landscape. Empirical results show that our method outperforms vanilla SVGD and a variety of baseline approaches over a range of real-world Bayesian inference tasks.

## 1 Introduction

Approximate inference of intractable distributions is a central task in probabilistic learning and statistics. An efficient approximation inference algorithm must perform both efficient *optimization* to explore the high probability regions of the distributions of interest, and reliable *uncertainty quantification* for evaluating the variation of the given distributions. Stein variational gradient descent (SVGD) (Liu & Wang, 2016) is a deterministic sampling algorithm that achieves both desiderata by optimizing the samples using a procedure similar to gradient-based optimization, while achieving reliable uncertainty estimation using an interacting repulsive mechanism. SVGD has been shown to provide a fast and flexible alternative to traditional methods such as Markov chain Monte Carlo (MCMC) (e.g., Neal et al., 2011; Hoffman & Gelman, 2014) and parametric variational inference (VI) (e.g., Wainwright et al., 2008; Blei et al., 2017) in various challenging applications (e.g., Pu et al., 2017; Wang & Liu, 2016; Kim et al., 2018; Haarnoja et al., 2017).

On the other hand, standard SVGD only uses the first order gradient information, and can not leverage the advantage of the second order methods, such as Newton's method and natural gradient, to achieve better performance on challenging problems with complex loss landscapes or domains. Unfortunately, due to the special form of SVGD, it is not straightforward to derive second order extensions of SVGD by simply extending similar ideas from optimization. While this problem has been recently considered (e.g., Detommaso et al., 2018; Liu & Zhu, 2018; Chen et al., 2019), the presented solutions either require heuristic approximations (Detommaso et al., 2018), or lead to complex algorithmic procedures that are difficult to implement in practice (Liu & Zhu, 2018).

Our solution to this problem is through a key generalization of SVGD that replaces the original scalar-valued positive definite kernels in SVGD with a class of more general *matrix-valued* positive definite kernels. Our generalization includes all previous variants of SVGD (e.g., Wang et al., 2018; Han & Liu, 2018) as special cases. More significantly, it allows us to easily incorporate various structured

---

[*]Equal contribution

preconditioning matrices into SVGD updates, including both Hessian and Fisher information matrices, as part of the generalized *matrix-valued* positive definite kernels. We develop theoretical results that shed insight on optimal design of the matrix-valued kernels, and also propose simple and fast practical procedures. We empirically evaluate both Newton and Fisher based extensions of SVGD on various practical benchmarks, including Bayesian neural regression and sentence classification, on which our methods show significant improvement over vanilla SVGD and other baseline approaches.

**Notation and Preliminary**    For notation, we use bold lower-case letters (e.g., $\boldsymbol{x}$) for vectors in $\mathbb{R}^d$, and bold upper-case letters (e.g., $\boldsymbol{Q}$) for matrices. A symmetric function $k\colon \mathbb{R}^d \times \mathbb{R}^d \to \mathbb{R}$ is called a positive definite kernel if $\sum_{ij} c_i k(\boldsymbol{x}_i, \boldsymbol{x}_j) c_j \geq 0$ for any $\{c_i\} \subset \mathbb{R}$ and $\{\boldsymbol{x}_i\} \subset \mathbb{R}^d$. Every positive definite kernel $k(\boldsymbol{x}, \boldsymbol{x}')$ is associated with a *reproducing kernel Hilbert space* (RKHS) $\mathcal{H}_k$, which consists of the closure of functions of form

$$f(\boldsymbol{x}) = \sum_i c_i k(\boldsymbol{x}, \boldsymbol{x}_i), \quad \forall \{c_i\} \subset \mathbb{R}, \ \{\boldsymbol{x}_i\} \subset \mathbb{R}^d, \tag{1}$$

for which the inner product and norm are defined by $\langle f, \ g \rangle_{\mathcal{H}_k} = \sum_{ij} c_i s_j k(\boldsymbol{x}_i, \boldsymbol{x}_j)$, $\|f\|_{\mathcal{H}_k}^2 = \sum_{ij} c_i c_j k(\boldsymbol{x}_i, \boldsymbol{x}_j)$, where we assume $g(\boldsymbol{x}) = \sum_i s_i k(\boldsymbol{x}, \boldsymbol{x}_i)$. Denote by $\mathcal{H}_k^d := \mathcal{H}_k \times \ldots \times \mathcal{H}_k$ the vector-valued RKHS consisting of $\mathbb{R}^d$ vector-valued functions $\boldsymbol{\phi} = [\phi^1, \ldots, \phi^d]^\top$ with each $\phi^\ell \in \mathcal{H}_k$. See e.g., Berlinet & Thomas-Agnan (2011) for more rigorous treatment. For notation convenience, we do not distinguish distributions on $\mathbb{R}^d$ and and their density functions.

## 2    Stein Variational Gradient Descent (SVGD)

We introduce the basic derivation of Stein variational gradient descent (SVGD), which provides a foundation for our new generalization. See Liu & Wang (2016, 2018); Liu (2017) for more details.

Let $p(\boldsymbol{x})$ be a positive and continuously differentiable probability density function on $\mathbb{R}^d$. Our goal is to find a set of points (a.k.a. particles) $\{\boldsymbol{x}_i\}_{i=1}^n \subset \mathbb{R}^d$ to approximate $p$, such that the empirical distribution $q(\boldsymbol{x}) = \sum_i \delta(\boldsymbol{x} - \boldsymbol{x}_i)/n$ of the particles weakly converges to $p$ when $n$ is large. Here $\delta(\cdot)$ denotes the Dirac delta function.

SVGD achieves this by starting from a set of initial particles, and iteratively updating them with a deterministic transformation of form

$$\boldsymbol{x}_i \leftarrow \boldsymbol{x}_i + \epsilon \boldsymbol{\phi}_k^*(\boldsymbol{x}_i), \quad \forall i = 1, \cdots, n, \qquad \boldsymbol{\phi}_k^* = \arg\max_{\boldsymbol{\phi} \in \mathcal{B}_k} \left\{ -\frac{\mathrm{d}}{\mathrm{d}\epsilon} \mathrm{KL}(q_{[\epsilon\boldsymbol{\phi}]} \,\|\, p) \Big|_{\epsilon=0} \right\}, \tag{2}$$

where $\epsilon$ is a small step size, $\boldsymbol{\phi}_k^*\colon \mathbb{R}^d \to \mathbb{R}^d$ is an optimal transform function chosen to maximize the decreasing rate of the KL divergence between the distribution of particles and the target $p$, and $q_{[\epsilon\boldsymbol{\phi}]}$ denotes the distribution of the updated particles $\boldsymbol{x}' = \boldsymbol{x} + \epsilon \boldsymbol{\phi}(\boldsymbol{x})$ as $\boldsymbol{x} \sim q$, and $\mathcal{B}_k$ is the unit ball of RKHS $\mathcal{H}_k^d := \mathcal{H}_k \times \ldots \times \mathcal{H}_k$ associated with a positive definite kernel $k(\boldsymbol{x}, \boldsymbol{x}')$, that is,

$$\mathcal{B}_k = \{\boldsymbol{\phi} \in \mathcal{H}_k^d\colon \ \|\boldsymbol{\phi}\|_{\mathcal{H}_k^d} \leq 1\}. \tag{3}$$

Liu & Wang (2016) showed that the objective in (2) can be expressed as a linear functional of $\boldsymbol{\phi}$,

$$-\frac{\mathrm{d}}{\mathrm{d}\epsilon} \mathrm{KL}(q_{[\epsilon\boldsymbol{\phi}]} \,\|\, p) \Big|_{\epsilon=0} = \mathbb{E}_{\boldsymbol{x} \sim q}[\mathcal{P}^\top \boldsymbol{\phi}(\boldsymbol{x})], \quad \mathcal{P}^\top \boldsymbol{\phi}(\boldsymbol{x}) = \nabla_{\boldsymbol{x}} \log p(\boldsymbol{x})^\top \boldsymbol{\phi}(\boldsymbol{x}) + \nabla_{\boldsymbol{x}}^\top \boldsymbol{\phi}(\boldsymbol{x}), \tag{4}$$

where $\mathcal{P}$ is a differential operator called *Stein operator*; here we formally view $\mathcal{P}$ and the derivative operator $\nabla_{\boldsymbol{x}}$ as $\mathbb{R}^d$ column vectors, hence $\mathcal{P}^\top \boldsymbol{\phi}$ and $\nabla_{\boldsymbol{x}}^\top \boldsymbol{\phi}$ are viewed as inner products, e.g., $\nabla_{\boldsymbol{x}}^\top \boldsymbol{\phi} = \sum_{\ell=1}^d \nabla_{x^\ell} \phi^\ell$, with $x^\ell$ and $\phi^\ell$ being the $\ell$-th coordinate of vector $\boldsymbol{x}$ and $\boldsymbol{\phi}$, respectively.

With (4), it is shown in Liu & Wang (2016) that the solution of (2) is

$$\boldsymbol{\phi}_k^*(\cdot) \propto \mathbb{E}_{\boldsymbol{x} \sim q}[\mathcal{P}k(\boldsymbol{x}, \cdot)] = \mathbb{E}_{\boldsymbol{x} \sim q}[\nabla_{\boldsymbol{x}} \log p(\boldsymbol{x}) k(\boldsymbol{x}, \cdot) + \nabla_{\boldsymbol{x}} k(\boldsymbol{x}, \cdot)]. \tag{5}$$

Such $\boldsymbol{\phi}_k^*$ provides the best update direction for the particles within RKHS $\mathcal{H}_k^d$. By taking $q$ to be the empirical measure of the particles, i.e., $q(\boldsymbol{x}) = \sum_{i=1}^n \delta(\boldsymbol{x} - \boldsymbol{x}_i)/n$ and repeatedly applying this update on the particles, we obtain the SVGD algorithm using equations (2) and (5).

# 3 SVGD with Matrix-valued Kernels

Our goal is to extend SVGD to allow efficient incorporation of precondition information for better optimization. We achieve this by providing a generalization of SVGD that leverages more general *matrix-valued* kernels, to flexibly incorporate preconditioning information.

The key idea is to observe that the standard SVGD searches for the optimal $\phi$ in RKHS $\mathcal{H}_k^d = \mathcal{H}_k \times \cdots \times \mathcal{H}_k$, a product of $d$ copies of RKHS of scalar-valued functions, which does not allow us to encode potential correlations between different coordinates of $\phi$. This limitation can be addressed by replacing $\mathcal{H}_k^d$ with a more general RKHS of vector-valued functions (called *vector-valued RKHS*), which uses more flexible *matrix-valued* positive definite kernels to specify rich correlation structures between different coordinates. In this section, we first introduce the background of vector-valued RKHS with matrix-valued kernels in Section 3.1, and then propose and discuss our generalization of SVGD using matrix-valued kernels in Section 3.2-3.3.

## 3.1 Vector-Valued RKHS with Matrix-Valued Kernels

We now introduce the background of matrix-valued positive definite kernels, which provides a most general framework for specifying vector-valued RKHS. We focus on the intuition and key ideas in our introduction, and refer the readers to Alvarez et al. (2012); Carmeli et al. (2006) for mathematical treatment.

Recall that a standard real-valued RKHS $\mathcal{H}_k$ consists of the closure of the linear span of its kernel function $k(\cdot, \boldsymbol{x})$, as shown in (1). Vector-valued RKHS can be defined in a similar way, but consist of the linear span of a *matrix-valued* kernel function:

$$\boldsymbol{f}(\boldsymbol{x}) = \sum_i \boldsymbol{K}(\boldsymbol{x}, \boldsymbol{x}_i)\boldsymbol{c}_i, \tag{6}$$

for any $\{\boldsymbol{c}_i\} \subset \mathbb{R}^d$ and $\{\boldsymbol{x}_i\} \subset \mathbb{R}^d$, where $\boldsymbol{K} \colon \mathbb{R}^d \times \mathbb{R}^d \to \mathbb{R}^{d \times d}$ is now a matrix-valued kernel function, and $\boldsymbol{c}_i$ are vector-valued weights. Similar to the scalar case, we can define an inner product structure $\langle \boldsymbol{f}, \boldsymbol{g} \rangle_{\mathcal{H}_{\boldsymbol{K}}} = \sum_{ij} \boldsymbol{c}_i^\top \boldsymbol{K}(\boldsymbol{x}_i, \boldsymbol{x}_j)\boldsymbol{s}_j$, where we assume $\boldsymbol{g} = \sum_i \boldsymbol{K}(\boldsymbol{x}, \boldsymbol{x}_i)\boldsymbol{s}_i$, and hence a norm $\|\boldsymbol{f}\|_{\mathcal{H}_k}^2 = \sum_{ij} \boldsymbol{c}_i^\top \boldsymbol{K}(\boldsymbol{x}_i, \boldsymbol{x}_j)\boldsymbol{c}_j$. In order to make the inner product and norm well defined, the matrix-value kernel $\boldsymbol{K}$ is required to be symmetric in that $\boldsymbol{K}(\boldsymbol{x}, \boldsymbol{x}') = \boldsymbol{K}(\boldsymbol{x}', \boldsymbol{x})^\top$, and positive definite in that $\sum_{ij} \boldsymbol{c}_i^\top \boldsymbol{K}(\boldsymbol{x}_i, \boldsymbol{x}_j)\boldsymbol{c}_j \geq 0$, for any $\{\boldsymbol{x}_i\} \subset \mathbb{R}^d$ and $\{\boldsymbol{c}_i\} \subset \mathbb{R}^d$.

Mathematically, one can show that the closure of the set of functions in (6), equipped with the inner product defined above, defines a RKHS that we denote by $\mathcal{H}_{\boldsymbol{K}}$. It is "reproducing" because it has the following reproducing property that generalizes the version for scalar-valued RKHS: for any $\boldsymbol{f} \in \mathcal{H}_{\boldsymbol{K}}$ and any $\boldsymbol{c} \in \mathbb{R}^d$, we have

$$\boldsymbol{f}(\boldsymbol{x})^\top \boldsymbol{c} = \langle \boldsymbol{f}(\cdot), \ \boldsymbol{K}(\cdot, \ \boldsymbol{x})\boldsymbol{c} \rangle_{\mathcal{H}_{\boldsymbol{K}}}, \tag{7}$$

where it is necessary to introduce $\boldsymbol{c}$ because the result of the inner product of two functions must be a scalar. A simple example of matrix kernel is $\boldsymbol{K}(\boldsymbol{x}, \boldsymbol{x}') = k(\boldsymbol{x}, \boldsymbol{x}')\boldsymbol{I}$, where $\boldsymbol{I}$ is the $d \times d$ identity matrix. It is related RKHS is $\mathcal{H}_{\boldsymbol{K}} = \mathcal{H}_k \times \cdots \times \mathcal{H}_k = \mathcal{H}_k^d$, as used in the original SVGD.

## 3.2 SVGD with Matrix-Valued Kernels

It is now natural to leverage matrix-valued kernels to obtain a generalization of SVGD (see Algorithm 1). The idea is simple: we now optimize $\phi$ in the unit ball of a general vector-valued RKHS $\mathcal{H}_{\boldsymbol{K}}$ with a matrix valued kernel $\boldsymbol{K}(\boldsymbol{x}, \boldsymbol{x}')$:

$$\phi_{\boldsymbol{K}}^* = \underset{\phi \in \mathcal{H}_{\boldsymbol{K}}}{\arg\max} \left\{ \mathbb{E}_{\boldsymbol{x} \sim q}\left[ \mathcal{P}^\top \phi(\boldsymbol{x}) \right], \ s.t. \ \|\phi\|_{\mathcal{H}_{\boldsymbol{K}}} \leq 1 \right\}. \tag{8}$$

This yields a simple closed form solution similar to (5).

**Theorem 1.** *Let $\boldsymbol{K}(\boldsymbol{x}, \boldsymbol{x}')$ be a matrix-valued positive definite kernel that is continuously differentiable on $\boldsymbol{x}$ and $\boldsymbol{x}'$, the optimal $\phi^*$ in (8) is*

$$\phi_{\boldsymbol{K}}^*(\cdot) \propto \mathbb{E}_{\boldsymbol{x} \sim q}\left[ \boldsymbol{K}(\cdot, \boldsymbol{x})\mathcal{P} \right] = \mathbb{E}_{\boldsymbol{x} \sim q}\left[ \boldsymbol{K}(\cdot, \boldsymbol{x})\nabla_{\boldsymbol{x}} \log p(\boldsymbol{x}) + \boldsymbol{K}(\cdot, \boldsymbol{x})\nabla_{\boldsymbol{x}} \right], \tag{9}$$

---

**Algorithm 1** Stein Variational Gradient Descent with Matrix-valued Kernels (Matrix SVGD)

---

**Input**: A (possibly unnormalized) differentiable density function $p(\boldsymbol{x})$ in $\mathbb{R}^d$. A matrix-valued positive definite kernel $\boldsymbol{K}(\boldsymbol{x}, \boldsymbol{x}')$. Step size $\epsilon$.

**Goal**: Find a set of particles $\{\boldsymbol{x}_i\}_{i=1}^n$ to represent the distribution $p$.

**Initialize** a set of particles $\{\boldsymbol{x}_i\}_{i=1}^n$, e.g., by drawing from some simple distribution.

**repeat**

$$\boldsymbol{x}_i \leftarrow \boldsymbol{x}_i + \frac{\epsilon}{n} \sum_{j=1}^n \left[ \boldsymbol{K}(\boldsymbol{x}_i, \boldsymbol{x}_j) \nabla_{\boldsymbol{x}_j} \log p(\boldsymbol{x}_j) + \boldsymbol{K}(\boldsymbol{x}_i, \boldsymbol{x}_j) \nabla_{\boldsymbol{x}_j} \right],$$

where $\boldsymbol{K}(\cdot, \boldsymbol{x}) \nabla_{\boldsymbol{x}}$ is formally defined as the product of matrix $\boldsymbol{K}(\cdot, \boldsymbol{x})$ and vector $\nabla_{\boldsymbol{x}}$. The $\ell$-th element of $\boldsymbol{K}(\cdot, \boldsymbol{x}) \nabla_{\boldsymbol{x}}$ is $(\boldsymbol{K}(\cdot, \boldsymbol{x}) \nabla_{\boldsymbol{x}})_\ell = \sum_{m=1}^d \nabla_{x^m} K_{\ell,m}(\cdot, \boldsymbol{x})$; see also (10).

**until** Convergence

---

*where the Stein operator $\mathcal{P}$ and derivative operator $\nabla_{\boldsymbol{x}}$ are again formally viewed as $\mathbb{R}^d$-valued column vectors, and $\boldsymbol{K}(\cdot, \boldsymbol{x})\mathcal{P}$ and $\boldsymbol{K}(\cdot, \boldsymbol{x})\nabla_{\boldsymbol{x}}$ are interpreted by the matrix multiplication rule. Therefore, $\boldsymbol{K}(\cdot, \boldsymbol{x})\mathcal{P}$ is a $\mathbb{R}^d$-valued column vector, whose $\ell$-th element is defined by*

$$(\boldsymbol{K}(\cdot, \boldsymbol{x})\mathcal{P})_\ell = \sum_{m=1}^d \left( K_{\ell,m}(\cdot, \boldsymbol{x}) \nabla_{x^m} \log p(\boldsymbol{x}) + \nabla_{x^m} K_{\ell,m}(\cdot, \boldsymbol{x}) \right), \qquad (10)$$

*where $K_{\ell,m}(\boldsymbol{x}, \boldsymbol{x}')$ denotes the $(\ell, m)$- element of matrix $\boldsymbol{K}(\boldsymbol{x}, \boldsymbol{x}')$ and $x^m$ the $m$-th element of $\boldsymbol{x}$.*

Similar to the case of standard SVGD, recursively applying the optimal transform $\phi_{\boldsymbol{K}}^*$ on the particles yields a general SVGD algorithm shown in Algorithm 1, which we call *matrix SVGD*.

Parallel to vanilla SVGD, the gradient of matrix SVGD in (9) consists of two parts that account for optimization and diversity, respectively: the first part is a weighted average of gradient $\nabla_{\boldsymbol{x}} \log p(\boldsymbol{x})$ multiplied by a matrix-value kernel $\boldsymbol{K}(\cdot, \boldsymbol{x})$; the other part consists of the gradient of the matrix-valued kernel $\boldsymbol{K}$, which, like standard SVGD, serves as a repulsive force to keep the particles away from each other to reflect the uncertainty captured in distribution $p$.

Matrix SVGD includes various previous variants of SVGD as special cases. The vanilla SVGD corresponds to the case when $\boldsymbol{K}(\boldsymbol{x}, \boldsymbol{x}') = k(\boldsymbol{x}, \boldsymbol{x}')\boldsymbol{I}$, with $\boldsymbol{I}$ as the $d \times d$ identity matrix; the gradient-free SVGD of Han & Liu (2018) can be treated as the case when $\boldsymbol{K}(\boldsymbol{x}, \boldsymbol{x}') = k(\boldsymbol{x}, \boldsymbol{x}')w(\boldsymbol{x})w(\boldsymbol{x}')\boldsymbol{I}$, where $w(\boldsymbol{x})$ is an importance weight function; the graphical SVGD of Wang et al. (2018); Zhuo et al. (2018) corresponds to a diagonal matrix-valued kernel: $\boldsymbol{K}(\boldsymbol{x}, \boldsymbol{x}') = \mathrm{diag}[\{k_\ell(\boldsymbol{x}, \boldsymbol{x}')\}_{\ell=1}^d]$, where each $k_\ell(\boldsymbol{x}, \boldsymbol{x}')$ is a "local" scalar-valued kernel function related to the $\ell$-th coordinate $x^\ell$ of vector $\boldsymbol{x}$.

### 3.3 Matrix-Valued Kernels and Change of Variables

It is well known that preconditioned gradient descent can be interpreted as applying standard gradient descent on a reparameterization of the variables. For example, let $\boldsymbol{y} = \boldsymbol{Q}^{1/2}\boldsymbol{x}$, where $\boldsymbol{Q}$ is a positive definite matrix, then $\log p(\boldsymbol{x}) = \log p(\boldsymbol{Q}^{-1/2}\boldsymbol{y})$. Applying gradient descent on $\boldsymbol{y}$ and transform it back to the updates on $\boldsymbol{x}$ yields a preconditioned gradient update $\boldsymbol{x} \leftarrow \boldsymbol{x} + \epsilon \boldsymbol{Q}^{-1} \nabla_{\boldsymbol{x}} \log p(\boldsymbol{x})$.

We now extend this idea to SVGD, for which matrix-valued kernels show up naturally as a consequence of change of variables. This justifies the use of matrix-valued kernels and provides guidance on the practical choice of matrix-valued kernels. We start with a basic result of how matrix-valued kernels change under change of variables (see Paulsen & Raghupathi (2016)).

**Lemma 2.** *Assume $\mathcal{H}_0$ is an RKHS with a matrix kernel $\boldsymbol{K}_0 \colon \mathbb{R}^d \times \mathbb{R}^d \to \mathbb{R}^{d \times d}$. Let $\mathcal{H}$ be the set of functions formed by*

$$\boldsymbol{\phi}(\boldsymbol{x}) = \boldsymbol{M}(\boldsymbol{x})\boldsymbol{\phi}_0(\boldsymbol{t}(\boldsymbol{x})), \qquad \forall \boldsymbol{\phi}_0 \in \mathcal{H}_0,$$

*where $\boldsymbol{M} \colon \mathbb{R}^d \to \mathbb{R}^{d \times d}$ is a fixed matrix-valued function and we assume $\boldsymbol{M}(\boldsymbol{x})$ is an invertible matrix for all $\boldsymbol{x}$, and $\boldsymbol{t} \colon \mathbb{R}^d \to \mathbb{R}^d$ is a fixed continuously differentiable one-to-one transform on $\mathbb{R}^d$.*

*For $\forall \boldsymbol{\phi}, \boldsymbol{\phi}' \in \mathcal{H}$, we can identity an unique $\boldsymbol{\phi}_0, \boldsymbol{\phi}_0' \in \mathcal{H}_0$ such that $\boldsymbol{\phi}(\boldsymbol{x}) = \boldsymbol{M}(\boldsymbol{x})\boldsymbol{\phi}_0(\boldsymbol{t}(\boldsymbol{x}))$ and $\boldsymbol{\phi}'(\boldsymbol{x}) = \boldsymbol{M}(\boldsymbol{x})\boldsymbol{\phi}_0'(\boldsymbol{t}(\boldsymbol{x}))$. Define the inner product on $\mathcal{H}$ via $\langle \boldsymbol{\phi}, \boldsymbol{\phi}' \rangle_{\mathcal{H}} = \langle \boldsymbol{\phi}_0, \boldsymbol{\phi}_0' \rangle_{\mathcal{H}_0}$, then $\mathcal{H}$ is*

*also a vector-valued RKHS, whose matrix-valued kernel is*

$$\boldsymbol{K}(\boldsymbol{x}, \boldsymbol{x}') = \boldsymbol{M}(\boldsymbol{x})\boldsymbol{K}_0(\boldsymbol{t}(\boldsymbol{x}), \boldsymbol{t}(\boldsymbol{x}'))\boldsymbol{M}(\boldsymbol{x}')^\top.$$

We now present a key result, which characterizes the change of kernels when we apply invertible variable transforms on the SVGD trajectory.

**Theorem 3.** *i) Let $p$ and $q$ be two distributions and $p_0$, $q_0$ the distribution of $\boldsymbol{x}_0 = \boldsymbol{t}(\boldsymbol{x})$ when $\boldsymbol{x}$ is drawn from $p$, $q$, respectively, where $\boldsymbol{t}$ is a continuous differentiable one-to-one map on $\mathbb{R}^d$. Assume $p$ is a continuous differentiable density with Stein operator $\mathcal{P}$, and $\mathcal{P}_0$ the Stein operator of $p_0$. We have*

$$\mathbb{E}_{\boldsymbol{x} \sim q_0}[\mathcal{P}_0^\top \boldsymbol{\phi}_0(\boldsymbol{x})] = \mathbb{E}_{\boldsymbol{x} \sim q}[\mathcal{P}^\top \boldsymbol{\phi}(\boldsymbol{x})], \qquad with \qquad \boldsymbol{\phi}(\boldsymbol{x}) := \nabla \boldsymbol{t}(\boldsymbol{x})^{-1} \boldsymbol{\phi}_0(\boldsymbol{t}(\boldsymbol{x})), \qquad (11)$$

*where $\nabla \boldsymbol{t}$ is the Jacobian matrix of $\boldsymbol{t}$.*

*ii) Therefore, in the asymptotics of infinitesimal step size ($\epsilon \to 0^+$), running SVGD with kernel $\boldsymbol{K}_0$ on $p_0$ is equivalent to running SVGD on $p$ with kernel*

$$\boldsymbol{K}(\boldsymbol{x}, \boldsymbol{x}') = \nabla \boldsymbol{t}(\boldsymbol{x})^{-1} \boldsymbol{K}_0(\boldsymbol{t}(\boldsymbol{x}), \boldsymbol{t}(\boldsymbol{x}')) \nabla \boldsymbol{t}(\boldsymbol{x}')^{-\top},$$

*in the sense that the trajectory of these two SVGD can be mapped to each other by the one-to-one map $\boldsymbol{t}$ (and its inverse).*

### 3.4 Practical Choice of Matrix-Valued Kernels

Theorem 3 suggests a conceptual procedure for constructing proper matrix kernels to incorporate desirable preconditioning information: one can construct a one-to-one map $\boldsymbol{t}$ so that the distribution $p_0$ of $\boldsymbol{x}_0 = \boldsymbol{t}(\boldsymbol{x})$ is an easy-to-sample distribution with a simpler kernel $\boldsymbol{K}_0(\boldsymbol{x}, \boldsymbol{x}')$, which can be a standard scalar-valued kernel or a simple diagonal kernel. Practical choices of $\boldsymbol{t}$ often involve rotating $\boldsymbol{x}$ with either Hessian matrix or Fisher information, allowing us to incorporating these information into SVGD. In the sequel, we first illustrate this idea for simple Gaussian cases and then discuss practical approaches for non-Gaussian cases.

**Constant Preconditioning Matrices** Consider the simple case when $p$ is multivariate Gaussian, e.g., $\log p(\boldsymbol{x}) = -\frac{1}{2}\boldsymbol{x}^\top \boldsymbol{Q}\boldsymbol{x} + const$, where $\boldsymbol{Q}$ is a positive definite matrix. In this case, the distribution $p_0$ of the transformed variable $\boldsymbol{t}(\boldsymbol{x}) = \boldsymbol{Q}^{1/2}\boldsymbol{x}$ is the standard Gaussian distribution that can be better approximated with a simpler kernel $\boldsymbol{K}_0(\boldsymbol{x}, \boldsymbol{x}')$, which can be chosen to be the standard RBF kernel suggested in Liu & Wang (2016), the graphical kernel suggested in Wang et al. (2018), or the linear kernels suggested in Liu & Wang (2018). Theorem 3 then suggests to use a kernel of form

$$\boldsymbol{K}_{\boldsymbol{Q}}(\boldsymbol{x}, \boldsymbol{x}') := \boldsymbol{Q}^{-1/2}\boldsymbol{K}_0\left(\boldsymbol{Q}^{1/2}\boldsymbol{x}, \ \boldsymbol{Q}^{1/2}\boldsymbol{x}'\right)\boldsymbol{Q}^{-1/2}, \qquad (12)$$

in which $\boldsymbol{Q}$ is applied on both the input $\boldsymbol{x}$ and the output side. As an example, taking $\boldsymbol{K}_0$ to be the scalar-valued Gaussian RBF kernel gives

$$\boldsymbol{K}_{\boldsymbol{Q}}(\boldsymbol{x}, \boldsymbol{x}') = \boldsymbol{Q}^{-1} \exp\left(-\frac{1}{2h}||\boldsymbol{x} - \boldsymbol{x}'||_{\boldsymbol{Q}}^2\right), \qquad (13)$$

where $||\boldsymbol{x} - \boldsymbol{x}'||_{\boldsymbol{Q}}^2 := (\boldsymbol{x} - \boldsymbol{x}')^\top \boldsymbol{Q}(\boldsymbol{x} - \boldsymbol{x}')$ and $h$ is a bandwidth parameter. Define $\boldsymbol{K}_{0,\boldsymbol{Q}}(\boldsymbol{x}, \boldsymbol{x}') := \boldsymbol{K}_0\left(\boldsymbol{Q}^{1/2}\boldsymbol{x}, \ \boldsymbol{Q}^{1/2}\boldsymbol{x}'\right)$. One can show that the SVGD direction of the kernel in (12) equals

$$\boldsymbol{\phi}_{\boldsymbol{K}_{\boldsymbol{Q}}}^*(\cdot) = \boldsymbol{Q}^{-1}\mathbb{E}_{\boldsymbol{x} \sim q}[\nabla \log p(\boldsymbol{x})\boldsymbol{K}_{0,\boldsymbol{Q}}(\cdot, \boldsymbol{x}) + \boldsymbol{K}_{0,\boldsymbol{Q}}(\cdot, \boldsymbol{x})\nabla_{\boldsymbol{x}}] = \boldsymbol{Q}^{-1}\boldsymbol{\phi}_{\boldsymbol{K}_{0,\boldsymbol{Q}}}^*(\cdot), \qquad (14)$$

which is a linear transform of the SVGD direction of kernel $\boldsymbol{K}_{0,\boldsymbol{Q}}(\boldsymbol{x}, \boldsymbol{x}')$ with matrix $\boldsymbol{Q}^{-1}$.

In practice, when $p$ is non-Gaussian, we can construct $\boldsymbol{Q}$ by taking averaging over the particles. For example, denote by $\boldsymbol{H}(\boldsymbol{x}) = -\nabla_{\boldsymbol{x}}^2 \log p(\boldsymbol{x})$ the negative Hessian matrix at $\boldsymbol{x}$, we can construct $\boldsymbol{Q}$ by

$$\boldsymbol{Q} = \sum_{i=1}^n \boldsymbol{H}(\boldsymbol{x}_i)/n, \qquad (15)$$

where $\{\boldsymbol{x}_i\}_{i=1}^n$ are the particles from the previous iteration. We may replace $\boldsymbol{H}$ with the Fisher information matrix to obtain a natural gradient like variant of SVGD.

**Point-wise Preconditioning**  A constant preconditioning matrix can not reflect different curvature or geometric information at different points. A simple heuristic to address this limitation is to replace the constant matrix $\boldsymbol{Q}$ with a point-wise matrix function $\boldsymbol{Q}(\boldsymbol{x})$; this motivates a kernel of form

$$\boldsymbol{K}(\boldsymbol{x}, \boldsymbol{x}') = \boldsymbol{Q}^{-1/2}(\boldsymbol{x}) \boldsymbol{K}_0\big(\boldsymbol{Q}^{1/2}(\boldsymbol{x})\boldsymbol{x}, \ \ \boldsymbol{Q}^{1/2}(\boldsymbol{x}')\boldsymbol{x}'\big) \boldsymbol{Q}^{-1/2}(\boldsymbol{x}').$$

Unfortunately, this choice may yield expensive computation and difficult implementation in practice, because the SVGD update involves taking the gradient of the kernel $\boldsymbol{K}(\boldsymbol{x}, \boldsymbol{x}')$, which would need to differentiate through matrix valued function $\boldsymbol{Q}(\boldsymbol{x})$. When $\boldsymbol{Q}(\boldsymbol{x})$ equals the Hessian matrix, for example, it involves taking the third order derivative of $\log p(\boldsymbol{x})$, yielding an unnatural algorithm.

**Mixture Preconditioning**  We instead propose a more practical approach to achieve point-wise preconditioning with a much simpler algorithm. The idea is to use a weighted combination of several constant preconditioning matrices. This involves leveraging a set of *anchor points* $\{\boldsymbol{z}_\ell\}_{\ell=1}^m \subset \mathbb{R}^d$, each of which is associated with a preconditioning matrix $\boldsymbol{Q}_\ell = \boldsymbol{Q}(\boldsymbol{z}_\ell)$ (e.g., their Hessian or Fisher information matrices). In practice, the anchor points $\{\boldsymbol{z}_\ell\}_{\ell=1}^m$ can be conveniently set to be the same as the particles $\{\boldsymbol{x}_i\}_{i=1}^n$. We then construct a kernel by

$$\boldsymbol{K}(\boldsymbol{x}, \boldsymbol{x}') = \sum_{\ell=1}^m \boldsymbol{K}_{\boldsymbol{Q}_\ell}(\boldsymbol{x}, \boldsymbol{x}') w_\ell(\boldsymbol{x}) w_\ell(\boldsymbol{x}'), \tag{16}$$

where $\boldsymbol{K}_{\boldsymbol{Q}_\ell}(\boldsymbol{x}, \boldsymbol{x}')$ is defined in (12) or (13), and $w_\ell(\boldsymbol{x})$ is a positive scalar-valued function that decides the contribution of kernel $\boldsymbol{K}_{\boldsymbol{Q}_\ell}$ at point $\boldsymbol{x}$. Here $w_\ell(\boldsymbol{x})$ should be viewed as a mixture probability, and hence should satisfy $\sum_{\ell=1}^m w_\ell(\boldsymbol{x}) = 1$ for all $\boldsymbol{x}$. In our empirical studies, we take $w_\ell(\boldsymbol{x})$ as the Gaussian mixture probability from the anchor points:

$$w_\ell(\boldsymbol{x}) = \frac{\mathcal{N}(\boldsymbol{x}; \boldsymbol{z}_\ell, \boldsymbol{Q}_\ell^{-1})}{\sum_{\ell'=1}^m \mathcal{N}(\boldsymbol{x}; \boldsymbol{z}_{\ell'}, \boldsymbol{Q}_{\ell'}^{-1})}, \qquad \mathcal{N}(\boldsymbol{x}; \boldsymbol{z}_\ell, \boldsymbol{Q}_\ell^{-1}) := \frac{1}{Z_\ell} \exp\left(-\frac{1}{2} \|\boldsymbol{x} - \boldsymbol{z}_\ell\|_{\boldsymbol{Q}_\ell}^2\right), \quad (17)$$

where $Z_\ell = (2\pi)^{d/2} \det(Q_\ell)^{-1/2}$. In this way, each point $\boldsymbol{x}$ is mostly influenced by the anchor point closest to it, allowing us to apply different preconditioning for different points. Importantly, the SVGD update direction related to the kernel in (16) has a simple and easy-to-implement form:

$$\boldsymbol{\phi}_{\boldsymbol{K}}^*(\cdot) = \sum_{\ell=1}^m w_\ell(\cdot) \mathbb{E}_{\boldsymbol{x} \sim q} \left[ (w_\ell(\boldsymbol{x}) \boldsymbol{K}_{\boldsymbol{Q}_\ell}(\cdot, \boldsymbol{x})) \mathcal{P} \right] = \sum_{\ell=1}^m w_\ell(\cdot) \boldsymbol{\phi}_{w_\ell \boldsymbol{K}_{\boldsymbol{Q}_\ell}}^*(\cdot), \tag{18}$$

which is a weighted sum of a number of SVGD directions with constant preconditioning matrices (but with an asymmetric kernel $w_\ell(\boldsymbol{x}) \boldsymbol{K}_{\boldsymbol{Q}_\ell}(\cdot, \boldsymbol{x})$).

**A Remark on Stein Variational Newton (SVN)**  Detommaso et al. (2018) provided a Newton-like variation of SVGD. It is motivated by an intractable functional Newton framework, and arrives a practical algorithm using a series of approximation steps. The update of SVN is

$$\boldsymbol{x}_i \leftarrow \boldsymbol{x}_i + \epsilon \tilde{\boldsymbol{H}}_i^{-1} \boldsymbol{\phi}_k^*(\boldsymbol{x}_i), \quad \forall i = 1, \dots, n, \tag{19}$$

where $\boldsymbol{\phi}_k^*(\cdot)$ is the standard SVGD gradient, and $\tilde{\boldsymbol{H}}_i$ is a Hessian like matrix associated with particle $\boldsymbol{x}_i$, defined by

$$\tilde{\boldsymbol{H}}_i = \mathbb{E}_{\boldsymbol{x} \sim q} \left[ \boldsymbol{H}(\boldsymbol{x}) k(\boldsymbol{x}, \boldsymbol{x}_i)^2 + (\nabla_{\boldsymbol{x}_i} k(\boldsymbol{x}, \boldsymbol{x}_i))^{\otimes 2} \right],$$

where $\boldsymbol{H}(\boldsymbol{x}) = -\nabla_{\boldsymbol{x}}^2 \log p(\boldsymbol{x})$, and $\boldsymbol{w}^{\otimes 2} := \boldsymbol{w}\boldsymbol{w}^\top$. Due to the approximation introduced in the derivation of SVN, it does not correspond to a standard functional gradient flow like SVGD (unless $\tilde{\boldsymbol{H}}_i = \boldsymbol{Q}$ for all $i$, in which case it reduces to using a constant preconditioning matrix on SVGD like (14)). SVN can be heuristically viewed as a "hard" variant of (18), which assigns each particle with its own preconditioning matrix with probability one, but the mathematical form do not match precisely. On the other hand, it is useful to note that the set of fixed points of SVN update (19) is the *identical to* that of the standard SVGD update with $\boldsymbol{\phi}_k^*(\cdot)$, once all $\tilde{\boldsymbol{H}}_i$ are positive definite matrices. This is because at the fixed points of (19), we have $\tilde{\boldsymbol{H}}_i^{-1} \boldsymbol{\phi}_k^*(\boldsymbol{x}_i) = 0$ for $\forall i = 1, \dots, n$, which is equivalent to $\boldsymbol{\phi}_k^*(\boldsymbol{x}_i) = 0, \forall i$ when all the $\tilde{\boldsymbol{H}}_i, \forall i$ are positive definite. Therefore, SVN can be justified as an alternative fixed point iteration method to achieve the same set of fixed points as the standard SVGD.

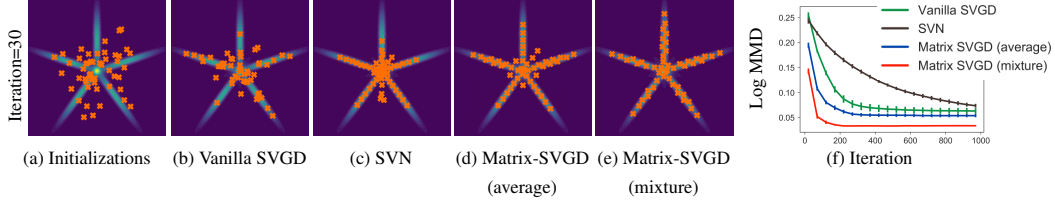

Figure 1: Figure (a)-(e) show the particles obtained by various methods at the 30-th iteration. Figure (f) plots the log MMD (Gretton et al., 2012) vs. training iteration starting from the 10-th iteration. We use the standard RBF kernel for evaluating MMD.

## 4 Experiments

We demonstrate the effectiveness of our matrix SVGD on various practical tasks. We start with a toy example and then proceed to more challenging tasks that involve logistic regression, neural networks and recurrent neural networks. For our method, we take the preconditioning matrices to be either Hessian or Fisher information matrices, depending on the application. For large scale Fisher matrices in (recurrent) neural networks, we leverage the Kronecker-factored (KFAC) approximation by Martens & Grosse (2015); Martens et al. (2018) to enable efficient computation. We use RBF kernel for vanilla SVGD. The kernel $\boldsymbol{K}_0(\boldsymbol{x}, \boldsymbol{x}')$ in our matrix SVGD (see (12) and (13)) is also taken to be Gaussian RBF. Following Liu & Wang (2016), we choose the bandwidth of the Gaussian RBF kernels using the standard median trick and use Adagrad (Duchi et al., 2011) for stepsize. Our code is available at `https://github.com/dilinwang820/matrix_svgd`.

The algorithms we test are summarized here:

`Vanilla SVGD`, using the code by Liu & Wang (2016);

`Matrix-SVGD (average)`, using the constant preconditioning matrix kernel in (13), with $\boldsymbol{Q}$ to be either the average of the Hessian matrices or Fisher matrices of the particles (e.g., (15));

`Matrix-SVGD (mixture)`, using the mixture preconditioning matrix kernel in (16), where we pick the anchor points to be particles themselves, that is, $\{\boldsymbol{z}_\ell\}_{\ell=1}^m = \{\boldsymbol{x}_i\}_{i=1}^n$;

`Stein variational Newton (SVN)`, based on the implementation of Detommaso et al. (2018);

`Preconditioned Stochastic Langevin Dynamics (pSGLD)`, which is a variant of SGLD (Li et al., 2016), using a diagonal approximation of Fisher information as the preconditioned matrix.

### 4.1 Two-Dimensional Toy Examples

**Settings** We start with illustrating our method using a Gaussian mixture toy model (Figure 1), with exact Hessian matrices for preconditioning. For fair comparison, we search the best learning rate for all algorithms exhaustively. We use 50 particles for all the cases. We use the same initialization for all methods with the same random seeds.

**Results** Figure 1 show the results for 2D toy examples. Appendix B shows more visualization and results on more examples. We can see that methods with Hessian information generally converge faster than vanilla SVGD, and `Matrix-SVGD (mixture)` yields the best performance.

### 4.2 Bayesian Logistic Regression

**Settings** We consider the Bayesian logistic regression model for binary classification. Let $\boldsymbol{D} = \{(\boldsymbol{x}_j, y_j)\}_{j=1}^N$ be a dataset with feature vector $\boldsymbol{x}_j$ and binary label $y_j \in \{0, 1\}$. The distribution of interest is

$$p(\boldsymbol{\theta} \mid \boldsymbol{D}) \propto p(\boldsymbol{D} \mid \boldsymbol{\theta})p(\boldsymbol{\theta}) \quad \text{with} \quad p(\boldsymbol{D} \mid \boldsymbol{\theta}) = \prod_{j=1}^N \left[ y_j \sigma(\boldsymbol{\theta}^\top \boldsymbol{x}_j) + (1 - y_j)\sigma(-\boldsymbol{\theta}^\top \boldsymbol{x}_j) \right],$$

where $\sigma(z) := 1/(1 + \exp(-z))$, and $p_0(\boldsymbol{\theta})$ is the prior distribution, which we set to be standard normal $\mathcal{N}(\boldsymbol{\theta}; \boldsymbol{0}, \boldsymbol{I})$. The goal is to approximate the posterior distribution $p(\boldsymbol{\theta} \mid \boldsymbol{D})$ with a set of

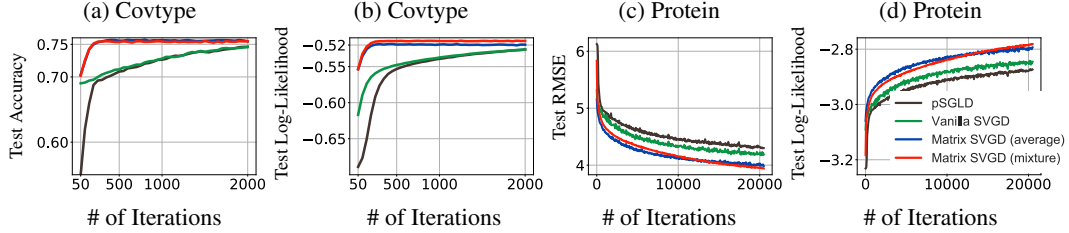

Figure 2: (a)-(b) Results of Bayesian Logistic regression on the Covtype dataset. (c)-(d) Average test RMSE and log-likelihood vs. training batches on the Protein dataset for Bayesian neural regression.

| Dataset | Test RMSE | | | | Test Log-Likelihood | | | |
|---|---|---|---|---|---|---|---|---|
| | pSGLD | Vanilla SVGD | Matrix-SVGD (average) | Matrix-SVGD (mixture) | pSGLD | Vanilla SVGD | Matrix-SVGD (average) | Matrix-SVGD (mixture) |
| Boston | **2.699±0.155** | 2.785±0.169 | 2.898±0.184 | 2.717±0.166 | −2.847±0.182 | −2.706±0.158 | **−2.669±0.141** | −2.861±0.207 |
| Concrete | 5.053±0.124 | 5.027±0.116 | 4.869±0.124 | **4.721±0.111** | −3.206±0.056 | **−3.064±0.034** | −3.150±0.054 | −3.207±0.071 |
| Energy | 0.985±0.024 | 0.889±0.024 | **0.795±0.025** | 0.868±0.025 | −1.395±0.029 | −1.315±0.020 | **−1.135±0.026** | −1.249±0.036 |
| Kin8nm | 0.091±0.001 | 0.093±0.001 | 0.092±0.001 | **0.090±0.001** | 0.973±0.010 | 0.964±0.012 | 0.956±0.011 | **0.975±0.011** |
| Naval | 0.002±0.000 | 0.004±0.000 | 0.001±0.000 | **0.000±0.000** | 4.535±0.093 | 4.312±0.087 | 5.383±0.081 | **5.639±0.048** |
| Combined | 4.042±0.034 | 4.088±0.033 | 4.056±0.033 | **4.029±0.033** | −2.821±0.009 | −2.832±0.009 | −2.824±0.009 | **−2.817±0.009** |
| Wine | 0.641±0.009 | 0.645±0.009 | **0.637±0.008** | 0.637±0.009 | −0.984±0.016 | −0.997±0.019 | **−0.980±0.016** | −0.988±0.018 |
| Protein | 4.300±0.018 | 4.186±0.017 | 3.997±0.018 | **3.852±0.014** | −2.874±0.004 | −2.846±0.003 | −2.796±0.004 | **−2.755±0.003** |
| Year | 8.630±0.007 | 8.686±0.010 | 8.637±0.005 | **8.594±0.009** | −3.568±0.002 | −3.577±0.002 | −3.569±0.001 | **−3.561±0.002** |

Table 1: Average test RMSE and log-likelihood in test data for UCI regression benchmarks.

particles $\{\boldsymbol{\theta}_i\}_{i=1}^n$, and then use it to predict the class labels for testing data points. We compare our methods with preconditioned stochastic gradient Langevin dynamics (pSGLD) (Li et al., 2016). Because pSGLD is a sequential algorithm, for fair comparison, we obtain the samples of pSGLD by running $n$ parallel chains of pSGLD for estimation. The preconditioning matrix in both pSGLD and matrix SVGD is taken to be the Fisher information matrix.

We consider the binary *Covtype*[2] dataset with $581,012$ data points and $54$ features. We partition the data into 70% for training, 10% for validation and 20% for testing. We use Adagrad optimizer with a mini-batch size of 256. We choose the best learning rate from $[0.001, 0.005, 0.01, 0.05, 0.1, 0.5, 1.0]$ for each method on the validation set. For all the experiments and algorithms, we use $n = 20$ particles. Results are average over 20 random trials.

**Results** Figure 2 (a) and (b) show the test accuracy and test log-likelihood of different algorithms. We can see that both `Matrix-SVGD (average)` and `Matrix-SVGD (mixture)` converge much faster than both vanilla SVGD and pSGLD, reaching an accuracy of 0.75 in less than 500 iterations.

### 4.3 Neural Network Regression

**Settings** We apply our matrix SVGD on Bayesian neural network regression on UCI datasets. For all experiments, we use a two-layer neural network with 50 hidden units with ReLU activation functions. We assign isotropic Gaussian priors to the neural network weights. All datasets[3] are randomly partitioned into $90\%$ for training and $10\%$ for testing. All results are averaged over 20 random trials, except for Protein and Year, on which 5 random trials are performed. We use $n = 10$ particles for all methods. We use Adam optimizer with a mini-batch size of 100; for large dataset such as *Year*, we set the mini-batch size to be 1000. We use the Fisher information matrix with Kronecker-factored (KFAC) approximation for preconditioning.

**Results** Table 1 shows the performance in terms of the test RMSE and the test log-likelihood. We can see that both `Matrix-SVGD (average)` and `Matrix-SVGD (mixture)`, which use second-order information, achieve better performance than vanilla SVGD. `Matrix-SVGD (mixture)` yields the best performance for both test RMSE and test log-likelihood in most cases. Figure 2 (c)-(d) show that both variants of Matrix-SVGD converge much faster than vanilla SVGD and pSGLD on the Protein dataset.

### 4.4 Sentence Classification With Recurrent Neural Networks (RNN)

**Settings** We consider the sentence classification task on four datasets: MR (Pang & Lee, 2005), CR (Hu & Liu, 2004), SUBJ (Pang & Lee, 2004), and MPQA (Wiebe et al., 2005). We use a recurrent neural network (RNN) based model, $p(y \mid \boldsymbol{x}) = \text{softmax}(\boldsymbol{w}_y^\top \boldsymbol{h}_{RNN}(\boldsymbol{x}, \boldsymbol{v}))$, where $\boldsymbol{x}$ is the input sentence, $y$ is a discrete-valued label of the sentence, and $\boldsymbol{w}_y$ is a weight coefficient related to label class $y$. And $\boldsymbol{h}_{RNN}(\boldsymbol{x}, \boldsymbol{v})$ is an RNN function with parameter $\boldsymbol{v}$ using a one-layer bidirectional GRU model (Cho et al., 2014) with 50 hidden units. We apply matrix SVGD to infer the posterior of $\boldsymbol{w} = \{\boldsymbol{w}_y \colon \forall y\}$, while updating the RNN

| Method | MR | CR | SUBJ | MPQA |
|---|---|---|---|---|
| SGLD | 20.52 | 18.65 | 7.66 | 11.24 |
| pSGLD | 19.75 | 17.50 | 6.99 | 10.80 |
| Vanilla SVGD | 19.73 | 18.07 | 6.67 | **10.58** |
| Matrix-SVGD (average) | 19.22 | 17.29 | 6.76 | 10.79 |
| Matrix-SVGD (mixture) | **19.09** | **17.13** | **6.59** | 10.71 |

Table 2: Sentence classification errors measured with four benchmarks.

weights $\boldsymbol{v}$ using typical stochastic gradient descent. In all experiments, we use the pre-processed text data provided in Gan et al. (2016). For all the datasets, we conduct 10-fold cross-validation for evaluation. We use $n = 10$ particles for all the methods. For training, we use a mini-batch size of 50 and run all the algorithms for 20 epochs with early stop. We use the Fisher information matrix for preconditioning.

**Results** Table 2 shows the results of testing classification errors. We can see that `Matrix-SVGD (mixture)` generally performs the best among all algorithms.

## 5 Conclusion

We present a generalization of SVGD by leveraging general matrix-valued positive definite kernels, which allows us to flexibly incorporate various preconditioning matrices, including Hessian and Fisher information matrices, to improve exploration in the probability landscape. We test our practical algorithms on various practical tasks and demonstrate its efficiency compared to various existing methods.

## Acknowledgement

This work is supported in part by NSF CRII 1830161 and NSF CAREER 1846421. We would like to acknowledge Google Cloud and Amazon Web Services (AWS) for their support.

## Footnotes

[2] https://www.csie.ntu.edu.tw/~cjlin/libsvmtools/datasets/binary.html

[3] https://archive.ics.uci.edu/ml/datasets.php

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
