[Supplementary Material · appendix.pdf]

# A Proof

**Proof of Theorem 1** Let $e_\ell$ be the column vector with 1 in $\ell^{th}$ coordinate and 0 elsewhere. By the RKHS reproducing property (7) we have

$$\mathbb{E}_{\boldsymbol{x} \sim q}\left[\mathcal{P}^\top \boldsymbol{\phi}(\boldsymbol{x})\right] = \mathbb{E}_{\boldsymbol{x} \sim q}\left[\nabla_{\boldsymbol{x}} \log p(\boldsymbol{x})^\top \boldsymbol{\phi}(\boldsymbol{x}) + \nabla_{\boldsymbol{x}}^\top \boldsymbol{\phi}(\boldsymbol{x})\right]$$

$$= \mathbb{E}_{\boldsymbol{x} \sim q}\left[\boldsymbol{\phi}(\boldsymbol{x})^\top \nabla_{\boldsymbol{x}} \log p(\boldsymbol{x}) + \sum_{\ell=1}^{d} \nabla_{x^\ell} \boldsymbol{\phi}(\boldsymbol{x})^\top e_\ell\right]$$

$$= \mathbb{E}_{\boldsymbol{x} \sim q}\left[\langle \boldsymbol{\phi}(\cdot), \ \boldsymbol{K}(\cdot, \boldsymbol{x}) \nabla_{\boldsymbol{x}} \log p(\boldsymbol{x})\rangle_{\mathcal{H}_{\boldsymbol{K}}} + \sum_{\ell=1}^{d} \nabla_{x^\ell} \langle \boldsymbol{\phi}(\cdot), \ \boldsymbol{K}(\cdot, \boldsymbol{x}) e_\ell\rangle_{\mathcal{H}_{\boldsymbol{K}}}\right]$$

$$= \left\langle \boldsymbol{\phi}(\cdot), \ \mathbb{E}_{\boldsymbol{x} \sim q}\left[\boldsymbol{K}(\cdot, \boldsymbol{x}) \nabla_{\boldsymbol{x}} \log p(\boldsymbol{x}) + \sum_{\ell=1}^{d} \nabla_{x^\ell} \boldsymbol{K}(\cdot, \boldsymbol{x}) e_\ell\right]\right\rangle_{\mathcal{H}_{\boldsymbol{K}}}$$

$$= \langle \boldsymbol{\phi}(\cdot), \ \mathbb{E}_{\boldsymbol{x} \sim q}\left[\boldsymbol{K}(\cdot, \boldsymbol{x}) \nabla_{\boldsymbol{x}} \log p(\boldsymbol{x}) + \boldsymbol{K}(\cdot, \boldsymbol{x}) \nabla_{\boldsymbol{x}}\right]\rangle_{\mathcal{H}_{\boldsymbol{K}}}$$

$$= \langle \boldsymbol{\phi}(\cdot), \ \mathbb{E}_{\boldsymbol{x} \sim q}\left[\boldsymbol{K}(\cdot, \boldsymbol{x}) \mathcal{P}\right]\rangle_{\mathcal{H}_{\boldsymbol{K}}},$$

The optimization in (8) is hence

$$\max_{\boldsymbol{\phi} \in \mathcal{H}_{\boldsymbol{K}}} \langle \boldsymbol{\phi}(\cdot), \ \mathbb{E}_{\boldsymbol{x} \sim q}\left[\boldsymbol{K}(\cdot, \boldsymbol{x}) \mathcal{P}\right]\rangle_{\mathcal{H}_{\boldsymbol{K}}}, \quad s.t. \quad \|\boldsymbol{\phi}\|_{\mathcal{H}_{\boldsymbol{K}}} \leq 1,$$

whose solution is $\boldsymbol{\phi}^*(\cdot) \propto \mathbb{E}_{\boldsymbol{x} \sim q}\left[\boldsymbol{K}(\cdot, \boldsymbol{x}) \mathcal{P}\right]$.

**Proof of Lemma 2** This is a basic result of RKHS, which can be found in classical textbooks such as Paulsen & Raghupathi (2016). The key idea is to show that $\boldsymbol{K}(\boldsymbol{x}, \boldsymbol{x}')$ satisfies the reproducing property for $\mathcal{H}$. Recall the reproducing property of $\mathcal{H}_0$:

$$\boldsymbol{\phi}_0(\boldsymbol{x})^\top \boldsymbol{c} = \langle \boldsymbol{\phi}_0, \ \boldsymbol{K}_0(\cdot, \boldsymbol{x}) \boldsymbol{c}\rangle_{\mathcal{H}_0}, \quad \forall \boldsymbol{c} \in \mathbb{R}^d.$$

Taking $\boldsymbol{\phi}(\boldsymbol{x}) = \boldsymbol{M}(\boldsymbol{x}) \boldsymbol{\phi}_0(\boldsymbol{t}(\boldsymbol{x}))$, we have

$$\boldsymbol{\phi}(\boldsymbol{x})^\top \boldsymbol{c} = \langle \boldsymbol{\phi}_0, \ \boldsymbol{K}_0(\cdot, \boldsymbol{t}(\boldsymbol{x})) \boldsymbol{M}(\boldsymbol{x})^\top \boldsymbol{c}\rangle_{\mathcal{H}_0}$$

$$= \langle \boldsymbol{\phi}, \ \boldsymbol{M}(\cdot) \boldsymbol{K}_0(\boldsymbol{t}(\cdot), \boldsymbol{t}(\boldsymbol{x})) \boldsymbol{M}(\boldsymbol{x})^\top \boldsymbol{c}\rangle_{\mathcal{H}}$$

$$= \langle \boldsymbol{\phi}, \ \boldsymbol{K}(\cdot, \boldsymbol{x}) \boldsymbol{c}\rangle_{\mathcal{H}},$$

where the second step follows $\langle \boldsymbol{\phi}, \boldsymbol{\phi}'\rangle_{\mathcal{H}} = \langle \boldsymbol{\phi}_0, \boldsymbol{\phi}_0'\rangle_{\mathcal{H}_0}$ with $\boldsymbol{\phi}_0'(\cdot) = \boldsymbol{K}_0(\cdot, \boldsymbol{t}(\boldsymbol{x})) \boldsymbol{M}(\boldsymbol{x})^T \boldsymbol{c}$.

**Proof of Theorem 3**

*Proof.* Note that KL divergence is invariant under invertible variable transforms, that is,

$$\text{KL}(q_{[\epsilon\phi]} \,||\, p) = \text{KL}(q_{[\epsilon\phi]0} \,||\, p_0). \tag{20}$$

where $p_0$ denotes the distribution of $\boldsymbol{x}_0 = \boldsymbol{t}(\boldsymbol{x})$ when $\boldsymbol{x} \sim p$, and $q_{[\epsilon\phi]0}$ denotes the distribution of $\boldsymbol{x}_0' = \boldsymbol{t}(\boldsymbol{x}')$ when $\boldsymbol{x}' \sim q_{[\epsilon\phi]}$. Recall that $q_{[\epsilon\phi]}$ is defined as the distribution of $\boldsymbol{x}' = \boldsymbol{x} + \epsilon\boldsymbol{\phi}(\boldsymbol{x})$ when $\boldsymbol{x} \sim q$.

Denote by $\boldsymbol{t}^{-1}$ the inverse map of $\boldsymbol{t}$, that is, $\boldsymbol{t}^{-1}(\boldsymbol{t}(\boldsymbol{x})) = \boldsymbol{x}$. We can see that $\boldsymbol{x}_0' \sim q_{[\epsilon\phi]0}$ can be obtained by

$$\boldsymbol{x}_0' = \boldsymbol{t}(\boldsymbol{x}') \qquad //\boldsymbol{x}' \sim q_{[\epsilon\phi]}$$

$$= \boldsymbol{t}(\boldsymbol{x} + \epsilon\boldsymbol{\phi}(\boldsymbol{x})) \qquad //\boldsymbol{x} \sim q$$

$$= \boldsymbol{t}(\boldsymbol{t}^{-1}(\boldsymbol{x}_0) + \epsilon\boldsymbol{\phi}(\boldsymbol{t}^{-1}(\boldsymbol{x}_0))) \qquad //\boldsymbol{x}_0 \sim q_0$$

$$= \boldsymbol{x}_0 + \epsilon\nabla\boldsymbol{t}(\boldsymbol{t}^{-1}(\boldsymbol{x}_0))\boldsymbol{\phi}(\boldsymbol{t}^{-1}(\boldsymbol{x}_0)) + \mathcal{O}(\epsilon^2)$$

$$= \boldsymbol{x}_0 + \epsilon\boldsymbol{\phi}_0(\boldsymbol{x}_0) + \mathcal{O}(\epsilon^2), \tag{21}$$

where we used the definition that $\boldsymbol{\phi}(\boldsymbol{x}) = \nabla\boldsymbol{t}(\boldsymbol{x})^{-1}\boldsymbol{\phi}_0(\boldsymbol{t}(\boldsymbol{x}))$ in (11), and $\mathcal{O}(\cdot)$ is the big-O notation.

From Theorem 3.1 of Liu & Wang (2016), we have

$$\frac{d}{d\epsilon}\text{KL}(q_{[\epsilon\phi]} \,||\, p)\Big|_{\epsilon=0} = -\mathbb{E}_q[\mathcal{P}^\top \phi].$$

Using Equation (21) and derivation similar to Theorem 3.1 of Liu & Wang (2016), we can show

$$\frac{d}{d\epsilon}\text{KL}(q_{[\epsilon\phi]0} \,||\, p_0)\Big|_{\epsilon=0} = -\mathbb{E}_{q_0}[\mathcal{P}_0^\top \phi_0].$$

Combining these with (20) proves (11).

Following Lemma 2, when $\phi_0$ is in $\mathcal{H}_0$ with kernel $\boldsymbol{K}_0(\boldsymbol{x}, \boldsymbol{x}')$, $\phi$ is in $\mathcal{H}$ with kernel $\boldsymbol{K}(\boldsymbol{x}, \boldsymbol{x}')$. Therefore, maximizing $\mathbb{E}_q[\mathcal{P}^\top \phi]$ in $\mathcal{H}$ is equivalent to $\mathbb{E}_{q_0}[\mathcal{P}_0^\top \phi_0]$ in $\mathcal{H}_0$. This suggests the trajectory of SVGD on $p_0$ with $\boldsymbol{K}_0$ and that on $p$ with $\boldsymbol{K}$ are equivalent.

$\square$

# B  Toy Examples

Figure 3 and Figure 4 show results of different algorithms on three 2D toy distributions: Star, Double banana and Sine. Detailed information of these distributions and more results are shown in Section B.1-B.3.

We can see from Figure 3-4 that both variants of matrix SVGD consistently outperform SVN and vanilla SVGD. We also find that `Matrix SVGD(mixture)` tends to outperform `Matrix SVGD (average)`, which is expected since `Matrix SVGD (average)` uses a constant preconditioning matrix for all the particles, and can not capture different curvatures at different locations. `Matrix SVGD (mixture)` yields the best performance in general.

Figure 3: The particles obtained by various methods at the 30/100/30-th iteration on three toy 2D distributions.

Figure 4: The MMD vs. training iteration of different algorithms on the three toy distributions.

## B.1 Sine

The density function of the "Sine" distribution is defined by

$$p(x_1, x_2) \propto \exp\left(\frac{-(x_2 + \sin(\alpha x_1))^2}{2\sigma_1} - \frac{x_1^2 + x_2^2}{2\sigma_2}\right),$$

where we choose $\alpha = 1$, $\sigma_1 = 0.003$, $\sigma_2 = 1$.

Figure 5: The particles obtained by various methods on the toy Sine distribution.

## B.2 Double Banana

We use the "double banana" distribution constructed in Detommaso et al. (2018), whose probability density function is

$$p(\boldsymbol{x}) \propto \exp\left(-\frac{\|\boldsymbol{x}\|_2^2}{2\sigma_1} - \frac{(y - F(\boldsymbol{x}))^2}{2\sigma_2}\right),$$

where $\boldsymbol{x} = [x_1, x_2] \in \mathbb{R}^2$ and $F(\boldsymbol{x}) = \log((1 - x_1)^2 + 100(x_2 - x_1^2)^2)$ and $y = \log(30)$, $\sigma_1 = 1.0, \sigma_2 = 0.09$.

Figure 6: The particles obtained by various methods on the double banana distribution.

## B.3 Star

We construct the "star" distribution with a Gaussian mixture model, whose density function is

$$p(\boldsymbol{x}) = \frac{1}{K} \sum_{i=1}^{K} \mathcal{N}(\boldsymbol{x}; \boldsymbol{\mu}_i, \boldsymbol{\Sigma}_i),$$

with $\boldsymbol{x} \in \mathbb{R}^2$ $\boldsymbol{\mu}_1 = [0; \ 1.5]$, $\boldsymbol{\Sigma}_1 = \mathrm{diag}([1; \ \frac{1}{100}])$, and the other means and covariance matrices are defined by rotating their previous mean and covariance matrix. To be precise,

$$\boldsymbol{\mu}_{i+1} = \boldsymbol{U}\boldsymbol{\mu}_i, \quad \boldsymbol{\Sigma}_{i+1} = \boldsymbol{U}\boldsymbol{\Sigma}_i\boldsymbol{U}^\top, \quad \boldsymbol{U} = \begin{bmatrix} \cos(\theta) & \sin(\theta) \\ -\sin(\theta) & \cos(\theta) \end{bmatrix},$$

with angle $\theta = \frac{2\pi}{K}$. We set the number of component $K$ to be 5.

Figure 7: The particles obtained by various methods on the star-shaped distribution.