[Reviews · NeurIPS 2019]

Reviewer 1



The paper is clearly written: both precise and easy to read. The main idea is very simple: remove the assumption that we equip the vector space of vector-valued functions H^d with the standard inner product (or equivalently standard kernel), i.e., take into account the vector components could be "correlated". While this is simple it does add clarity to the theoretical setting of SVGD in which it is assumed, but not stated, that the components are independent. As the authors show it is important to note H^d is a vector-valued RKHS since coordinate transformation (which should not affect experiments) induce non-standard matrix kernels. While this provides the natural theoretical framework, it is raises the question of which matrix kernel to choose and I think the authors do not really answer this important (but probably hard) question. Indeed it seems to me that the natural way to choose the matrix kernel is using some intrinsic geometric information (coordinate independent), but the authors explain themselves that this leads to expensive computation. It is not clear to me why the "mixture preconditioning kernel" would work better in general.

Reviewer 2



*** UPDATE *** Thank you for your response. This has clarified some of my comments, and thus I have increased my score by one level. However, a negative aspect from the author response was the excuse given for the absence of a comparison against SVN; "the results we obtained were much worse than our methods and baselines, and hence did not investigate it further". To me this seems like a very obvious thing to include and discuss in the paper, lending strong support to the new method, so I am a bit suspicious about why it wasn't included. I hope the authors will include it in the revised manuscript, if accepted. Another reviewer commented on the lack of wall-time comparison, and again I feel that the authors did not give a valid excuse for omitting this information from the manuscript (the reader can, I think, be trusted to adjust for different computational setups and hardware when interpreting the wall-time data). This paper generalises the Stein variational gradient descent method to the case of a matrix-valued kernel. Various matrix-valued kernels are proposed in this context and they are empirically studied. The idea is novel and the method is interesting enough. The main criticisms I have about this work are the limited nature of both the empirical assessment and the description of the empirical assessment itself. In particular, what I would consider to be some key empirical comparisons haven't been included and key implementational details needed to interpret the results aren't provided. Code is provided, but the reader should not be expected to reverse-engineer the experiments from the script. In this respect, a minimal standard for reproducible scientific writing has not been met. l14. The authors claim that distributional approximation is somehow more difficult than optimisation. This discussion seems fluffy and not well-defined, since of course SVGD is also an optimisation method. I would just remove l14-16. l39. "extends" -> "extensions" l55. The authors should remove the word "complex". l61. "unite" -> "unit" l61. "the" unit ball is not unique - it depends on the choice of norm on R^d. The norm on R^d being used should be stated. The absence of precision on this point has the consequence that it is also unclear how ||.||_{H_k^d} is being defined. l120. The authors mention that "vanilla SVGD" is a special case of their method. Is R-SVGD a special case of their method? If not, this should be acknowledged. l130. I may be wrong, but should the "+" be a "-"? Otherwise it looks like gradient ascent instead of descent. l152. The authors could note that the use of constant preconditioning matrices in the context of kernels and Stein's method was considered also in [CBBGGMO2019]. l207. A reference is needed for the median trick. l210. The authors write that in the experiments Q was "either" the average Hessian or the Fisher information matrix of the particles. This is not good enough - the authors need to be precise about what was used for what experiment. l211. For each experiment, the number of particles / anchor points used to form the mixture preconditioning matrix kernel in Matrix-SVGD is not stated. It is also not stated how the anchor points were selected. The effect of these factors on the results was not explored. As such, I cannot properly interpret any of the empirical results for Matrix-SVGD. l216. The authors need to state what kernel was used to compute the MMD reported in Fig. 1. (This should not be the preconditioner kernel used for Matrix-SVGD, as this would give the proposed method(s) an unfair advantage.) l216. It appears that competing methods may not be initialised from the same initial point set. Indeed, in Fig. 1(f) very different values of MMD are reported when the number of iterations is small. To avoid doubt, the authors should carefully say what initial configuration(s) of point sets were used. l223. Why was SVNM (probably the most closely related existing method to the Matrix-SVGD method being proposed) not included in any of the three "real" empirical experiments (4.2, 4.3, 4.4)? l240. The authors do not state whether mini-batching was used in 4.3 and 4.4. If it was used, it should be stated, and the corresponding implementational details provided. l240. Neither the number of iterations, nor the learning rate are provided for 4.3 nor 4.4. This is basic and important information, and I can't understand why this was not included anywhere in the manuscript or supplement. lA339. The interchange of expectation and inner product should be justified. What assumption(s) are required? lA340. The authors just "cite a book" in the proof of Lemma 2, whereas a pointer to the specific result in the book (and checking of any preconditions) is needed. [CBBGGMO2019] Chen WY, Barp A, Briol FX, Gorham J, Girolami M, Mackey L, Oates CJ. Stein Point Markov Chain Monte Carlo. International Conference on Machine Learning (ICML 2019).

Reviewer 3



The paper provides a nice generalization of SVGD. It does this by optimizing the KSD over a different unit ball of functions--instead of using the direct sum of d copies of an RKHS with scalar-valued kernel, it considers the unit ball of vector valued functions that arise from considering an RKHS associated with a matrix valued kernel. This allows one to better capture local information in the density functions, as evidenced in the experiments. The authors also point out how this variant of SVGD reduces to many other variants of SVGD as a special case. L102: There looks to be a typo. Do you mean to say the vv-RKHS is equivalent to the direct sum of d copies of H_k? Experiments 1 & 2: Is there any way to compare the algorithms on CPU time rather than iterations? The matrix valued SVGD will take longer per iteration, and it would be nice to see how these approaches compare as a function of time. Originality: The paper produces a generalization of SVGD that captures many other already known variants. It also provides a computationally feasible version that is useful in the experiments shown. Quality: The paper appears to be correct. Clarity: The paper is very well written and easy to follow. It was a nice read. Significance: The paper is a nice contribution to the topic of SVGD and provides some ideas for incorporating more information about the score function.

[Author Response · NeurIPS 2019]

**[Reviewer #1]** Thank you for your valuable comments and suggestions.

*On choosing a matrix kernel in practice*: As the reviewer correctly noted, the problem of choosing an optimal kernel is
a very hard problem. The difficulty lies on the fact that ideal optimal kernel should incorporate the intrinsic geometric
information, but leads to no expensive computation.

In this work, we propose the mixture preconditing kernel as a practical heuristic to strike the balance between
efficiency and computational cost. Compared with the constant preconditioning (which corresponds to using constant
Hessian matrix), the mixture preconditioning allows us to incorporate different curvatures in different locations, which
is of critical importance for complex models such as the "star" in Figure 1. Meanwhile, compared with the more
theoretically motivated kernel derived with change of variables, the mixture preconditining kernel avoids taking high
order derivatives and is much more computationally efficiently.

**[Reviewer #2]** Thank you for your valuable comments. We will improve our presentation based on your questions.

*Comparisons with SVNM* : The advantage of matrix SVGD over SVN is clear both theoretically and empirically. As we
point out in L188-199, SVN can be in fact viewed as solving the same fixed point equation of SVGD with a scalar
kernel and hence can not leverage the advantage of matrix kernels. For empirical comparison on the realistic examples,
we did test the official Matlab code provide by the SVN authors, but the results we obtained were much worse than our
methods and baselines (similar to the results in Fig 1), and hence did not investigate it further. For example, the RMSE
score of SVN of BNN on the Boston dataset is $3.1830(\pm0.1829)$ (averaged on 20 random trials), which is much worse
than all the methods reported in Table 1 ($< 2.9$). We will investigate further and add discussion on this issue.

*The unit ball is not unique*: The RKHS $\mathcal{H}_k^d := \mathcal{H}_k \times \cdots \times \mathcal{H}_k$ must be equipped with inner product $\langle f, g \rangle_{\mathcal{H}_k^d} =$
$\sum_{i=1}^{d} \langle f_i,\ g_i \rangle_{\mathcal{H}_k}$, which corresponds to using $L_2$ norm on $\mathbb{R}^d$. Using other norms on $\mathbb{R}^d$ can not define an RKHS.

*Is R-SVGD a special case of our method*: No, we will acknowledge out this.

*[CBBGGMO2019]* : Thanks for pointing it out. We were not aware this work by the time of submission. We will add a
reference. But note that the preconditioned kernel in this paper is still a scalar kernel and hence substantially different
from our matrix kernel.

*Preconditioning matrices*: We use Hessian matrix in Sec 4.1 and Fisher with K-FAC approximation in Sec 4.2-4.4.

*How many particles*: For the toy example, we use 50 particles for all the cases. For other applications, please see *line
235*, *line 245* and *line 264* for more details.

*How many anchor points* : Please see *line 212* (we use all the particles as the anchor points).

*What kernel is used for computing MMD in Fig 1*: We use the standard RBF kernel for evaluating MMD.

*Different initialization?*: We did use the same initialization for all methods with the same random seeds. Please see Fig
5, 6, 7; or code/bnn/trainer.py:L293, or code/rnn/acl_setence_classification.py: L384; in Figure 1(f), the plot is started
from the 10-th iteration to avoid large values for better visualization.

*Section 4.3 and 4.4: mini-batch size and #iteration* : For Section 4.3, we use a mini-batch size of 100 for training; for
large dataset such as year, we set the batch size to be 1000. In general, we grid search the least training epochs required
for our implementation of SVGD to have a performance that matches the results reported in the original SVGD paper;
we then train all other methods with the same training epochs. For Section 4.4, we use a mini batch-size of 50. we train
all algorithms for 20 epochs;

*the interchange of expectation and inner product should be justified*: It is true once integrations involved are finite.

*Reference*: We will add a reference on the median trick and Lemma 2.

**[Reviewer #3]** Thank you for your valuable comments and suggestions.

*L102: There looks to be a typo*: Yes, $\mathcal{H}_k^d$ should be corrected to be $\mathcal{H}_{\mathbf{K}}$ here.

*Experiments 1 & 2, is there any way to compute the algorithms on CPU time rather than iterations* : As the other cases
when Newton or natural gradient are used, the timing comparison heavily depends on implementation, approximation
method of the matrix inversion, and many other factors such as matlab or python code and gpu-based or cpu-based
implementation. Another important point, unique to our method, is that the SVGD with matrix kernels can yield *a set of
better fixed points* than SVGD with scalar kernels, which may worth the effort even when the time-wise convergence is
slower (in comparison, the standard Newton's method and natural gradient yield the same set of fixed point as standard
gradient descent).

[Meta-Review · NeurIPS 2019]

The reviewers felt that this submission represents an important contribution to the field. Please be sure to carefully review and address the concerns of all reviewers (e.g., introducing a comparison with SVNM and providing complete implementation details) in the revision.